# Sensitivity study of cloud parameterizations with relative dispersion in CAM5.1: impacts on aerosol indirect effects

Xiaoning Xie[1], He Zhang[2], Xiaodong Liu[1,3], Yiran Peng[4], and Yangang Liu[5]

[1]SKLLQG, Institute of Earth Environment, Chinese Academy of Sciences, Xi'an 710061, China
[2]International Center for Climate and Environment Sciences, Institute of Atmospheric Physics, Chinese Academy of Sciences, Beijing 100029, China
[3]University of Chinese Academy of Sciences, Beijing 100049, China
[4]Ministry of Education Key Laboratory for Earth System Modeling, Center for Earth System Science, and Joint Center for Global Change Studies (JCGCS), Tsinghua University, Beijing 100084, China
[5]Environmental and Climate Sciences Department, Brookhaven National Laboratory, Upton, NY 11973-5000, USA

*Correspondence to:* Xiaoning Xie (xnxie@ieecas.cn)

**Abstract.** Aerosol-induced increase of relative dispersion of cloud droplet size distribution $\varepsilon$ exerts a warming effect and partly offsets the cooling of aerosol indirect radiative forcing (AIF) associated with increased droplet concentration by increasing the cloud droplet effective radius ($R_e$) and enhancing the cloud-to-rain autoconversion rate ($Au$) (labeled as dispersion effect), which can help reconcile global climate models (GCMs) with the satellite observations. However, the total dispersion effects

5    on both $R_e$ and $Au$ are not fully considered in most GCMs, especially in different versions of the Community Atmospheric Model (CAM). In order to accurately evaluate the dispersion effect on AIF, the new complete cloud parameterizations of $R_e$ and $Au$ explicitly accounting for $\varepsilon$ are implemented into the CAM version 5.1 (CAM5.1), and a suite of sensitivity experiments is conducted with different representations of $\varepsilon$ reported in literature. It is shown that the shortwave cloud radiative forcing is much better simulated with the new cloud parameterizations as compared to the standard scheme in CAM5.1, whereas the

10    influences on longwave cloud radiative forcing and surface precipitation are minimal. Additionally, consideration of dispersion effect can significantly reduce the changes induced by anthropogenic aerosols in the cloud top effective radius and the liquid water path, especially in Northern Hemisphere. The corresponding AIF with dispersion effect considered can also be reduced substantially, by a range of 0.10 to 0.21 W m$^{-2}$ at global scale, and by a much bigger margin of 0.25 to 0.39 W m$^{-2}$ for the Northern Hemisphere in comparison with that fixed relative dispersion, mainly dependent on the change of relative dispersion

15    and droplet concentrations ($\Delta\varepsilon/\Delta N_c$).

## 1 Introduction

It is well known that anthropogenic aerosols serving as cloud condensation nuclei (CCN) can enhance the cloud droplet concentration and decrease the droplet size, thereby increasing the cloud albedo for a given liquid water content (Twomey, 1977), as well as lifetime and coverage of clouds (Albrecht, 1989). Despite much attention and effort over the last few decades

20    (Ramanathan et al. 2001; Lohmann and Feichter, 2005), the so-called first and second aerosol indirect effects continue to suffer from large uncertainties in climate models (IPCC, 2007; IPCC, 2013).

Key to climate model estimates of the aerosol indirect radiative forcing (AIF) are the parameterizations of the cloud droplet effective radius ($R_e$) and the cloud-to-rain autoconversion rate ($Au$), which affect the first and second aerosol indirect effects, respectively. The $R_e$, which is defined as the ratio of the third to the second moment based on the cloud droplet size distribution, is one of the key variables that are used for calculating radiative properties of liquid water clouds. The decrease in $R_e$ due to the increased droplet concentration induced by increased aerosols can increase the cloud optical depth, the cloud albedo, and in turn enhance the cloud radiative forcing (Twomey, 1977). Additionally, the $Au$ process represents a key microphysical process linking cloud droplets formed by the diffusional growth and raindrops formed by the collision/coalescence processes in warm clouds. Note that this microphysical process of $Au$ is an important player in aerosol loadings, cloud morphology, and precipitation processes because that changes induced by aerosols in cloud microphysical properties can affect the spatio-temporal rainfall variations in addition to the onset and amount of rainfall. A lower efficiency of the $Au$ process resulting form increased aerosols can reduce the precipitation efficiency, prolong the cloud lifetime, and also enhance the cloud radiative forcing (Albrecht, 1989). Hence, improving parameterizations of $R_e$ and $Au$ are expected to significantly reduce the uncertainty of the first and second indirect aerosol effects, and further advance the scientific understanding of aerosol-cloud-radiation-precipitation-climate interactions (Liu and Daum, 2002; Liu and Daum, 2004; Guo et al, 2008; Liu and Li, 2015; Xie and Liu, 2015).

It is well established that effective radius (Martin et al; 1994; Liu and Daum, 2002) and autoconversion rate (Liu and Daum, 2004; Liu et al., 2007; Xie and Liu, 2009; Li et al., 2008; Chuang et al. 2012; Wang et al., 2013; Michibata and Takemura, 2015) are both related to the relative dispersion of cloud droplet size distribution $\varepsilon$ (which is defined as the ratio of the standard deviation to the mean value of droplet size distribution) in addition to droplet number concentration and cloud liquid water content. Liu and Daum (2002) suggested that $\varepsilon$ is increased by anthropogenic aerosols under similar dynamical conditions in clouds, because more numerous small droplets formed in polluted clouds compete for water vapor and broaden the droplet size distribution compared with clean clouds having fewer droplets and less competition. Further theoretical study (Liu et al., 2006) revealed that the increased $\varepsilon$ is primarily due to slowdown of condensational narrowing associated with decreased supersaturation. The enhanced $\varepsilon$ can increase effective radius and autoconversion rate, and thus exert a warming effect, offsetting the first and second aerosol indirect effects caused by the aerosol-induced change in droplet concentration, and helping reduce the uncertainty and discrepancy between climate model estimates and satellite observations. Furthermore, they estimated that the dispersion effect may reduce the magnitude of the first aerosol indirect effect by $10-80\%$ depending on the parameterization of $\varepsilon$. However, only few GCM studies (e. g., ECHAM4; CSIRO Mark3 GCM) in literature have either considered the dispersion effect on $R_e$ (Peng and Lohmann, 2003; Rotstayn and Liu, 2003; Rotstayn and Liu, 2009) or use the parameterization of $Au$ with $\varepsilon$ in mass content (Rotstayn and Liu, 2005). There has been no comprehensive investigation to examine the dispersion effect through both effective radius and autoconversion process with two-moment schemes. Although the microphysical scheme of Community Atmospheric Model version 5.1 (CAM5.1) considers dispersion effect on the cloud droplet effective radius (Morrison and Gettelman, 2008), it uses an expression different from other studies and no systematic examination of the influence of using different expressions on the model results. Furthermore, it is noted that the CAM5.1 microphysical scheme does not consider dispersion effect on the cloud-to-rain autoconversion process. To address the dispersion effect in

CAM5.1, we first implement the complete cloud microphysical parameterizations of $R_e$ and the two-moment $Au$ with $\varepsilon$ based on the Gamma size distribution function into CAM5.1. This new implementation allows us to address the dispersion effects on CAM5.1 simulations in general and the first and second aerosol indirect radiative forcing in particular.

The rest part of this paper is organized as follows. Section 2 describes of the microphysical parameterizations of $R_e$ and the two-moment $Au$ with $\varepsilon$ based on the Gamma size distribution function, as well as the parameterization of $\varepsilon$. Section 3 presents the description of CAM5.1 and evaluate the simulated cloud fields and precipitation with the new cloud microphysical parameterizations against observations. In Sect. 4, we investigate the dispersion effects on $R_e$ and $Au$, and furthermore on AIF. Finally, the main results are summarized in Sect. 5.

## 2   Descriptions of parameterizations of effective radius, autoconversion process, and relative dispersion

Most bulk cloud microphysical schemes in climate models are based upon the assumption that the cloud droplet size distribution can be represented by a Gamma size distribution

$$n(r) = \frac{N_c \lambda^{\mu+1}}{\Gamma(\mu+1)} r^\mu exp(-\lambda r), \tag{1}$$

where $r$ is the radius of a cloud droplet, $n(r)$ is the cloud droplet number concentration per unit of droplet radius interval $r$, $N_c$ is the cloud droplet number concentration, $\lambda$ is the slope parameter, and $\mu$ is the shape parameter related to $\varepsilon$ ($\varepsilon = (\mu+1)^{-1/2}$). The corresponding Gamma function is defined as $\Gamma(n) = \int_0^\infty x^{n-1} e^{-x} dx$, and the incomplete Gamma function is $\Gamma(n,a) = \int_a^\infty x^{n-1} e^{-x} dx$.

For the Gamma droplet size distribution (1), the cloud droplet effective radius $R_e$ can be parameterized via the following expression (Liu and Daum, 2000; Liu and Daum, 2002)

$$R_e = \frac{\int_0^\infty r^3 n(r) dr}{\int_0^\infty r^2 n(r) dr} = (\frac{3}{4\pi\rho_w})^{1/3} \beta(\varepsilon) (\frac{L_c}{N_c})^{1/3}, \tag{2}$$

where the microphysical properties $N_c$ and $L_c$ represent the droplet number concentration and the cloud liquid water content, respectively; and the variable $\rho_w$ is water density; the the effective radius ratio $\beta(\varepsilon)$ is a function of $\varepsilon$ described by $\beta(\varepsilon) = \frac{(1+2\varepsilon^2)^{2/3}}{(1+\varepsilon^2)^{1/3}}$. Note that this theoretical parameterization about $R_e$ is similar to that in CAM5.1 (Morrison and Gettelman 2008), except that it is directly related to the parameter $\varepsilon$ through Eq. (2). This explicit relationship permits a direct investigation of the dispersion influence on the first aerosol indirect effect.

According to the generalized mean value theorem for integrals (Liu and Daum, 2004; Liu et al., 2007), the two-moment parameterizations of $Au$ can be easily derived based upon the equation of the Gamma droplet size distribution from the results of Xie and Liu (2009),

$$P_N = 1.1 \times 10^{10} \frac{\Gamma(\varepsilon^{-2}, x_{cq})\Gamma(\varepsilon^{-2}+6, x_{cq})}{\Gamma^2(\varepsilon^{-2}+3)} L_c{}^2,$$

$$P_L = 1.1 \times 10^{10} \frac{\Gamma(\varepsilon^{-2})\Gamma(\varepsilon^{-2}+3, x_{cq})\Gamma(\varepsilon^{-2}+6, x_{cq})}{\Gamma^3(\varepsilon^{-2}+3)} N_c{}^{-1} L_c{}^3, \tag{3}$$

where $P_N$ (cm$^{-3}$ s$^{-1}$) and $P_L$ (g cm$^{-3}$ s$^{-1}$) are the autoconversion rates for cloud droplet number concentration and mass content, respectively. The parameter $x_{cq}$ can be written as a formula $x_{cq} = [\frac{(1+2\varepsilon^2)(1+\varepsilon^2)}{\varepsilon^4}]^{1/3} x_c^{1/3}$ where $x_c = 9.7 \times 10^{-17} N_c^{3/2} L_c^{-2}$. $P_N$ and $P_L$ are the increasing functions of $L_c$ and $\varepsilon$, as well as the decreasing functions of $N_c$ (Liu et al., 2007; Xie and Liu, 2009). This two-moment parameterization of $Au$ that explicitly accounts for $\varepsilon$ is used to replace the KK parameterization in the original CAM5.1 (Khairoutdinov and Kogan, 2000) to investigate the impact of $\varepsilon$ on the second aerosol indirect effect.

Several empirical expressions have been proposed to represent $\varepsilon$ in terms of the droplet number concentration (reviewed by Xie et al. 2013). Here, three commonly used expressions are used to investigate the dispersion effect. The Morrison-Grabowski relationship is given by Morrison and Grabowski (2007) based on the observational data from warm stratocumulus clouds (Martin et al., 1994)

$$\varepsilon = 0.0005714 N_c + 0.271. \tag{4}$$

Based on the observational data derived from Liu and Daum (2000), the Rotstayn-Liu relationship is presented as the following analytical formulation by Rotstayn and Liu (2003)

$$\varepsilon = 1 - 0.7 exp(-\alpha N_c), \tag{5}$$

where the constant $\alpha = 0.001$, 0.003 or 0.008, and here we adopt the value of $\alpha = 0.003$ which is more reasonable in global simulation as suggested by Rotstayn and Liu (2003). Note that the Morrison-Grabowski relationship has been used in the CAM5.1 microphysics scheme (Neale et al., 2010), and the Rotstayn-Liu relationship is coupled to the corresponding microphysics scheme of the CSIRO Mark3 GCM as described by Rotstayn and Liu (2003).

It is noted that the above two expressions both relate $\varepsilon$ to droplet concentration and ignore the influence of varying liquid water content. Wood (2000) showed that the effective radius ratio $\beta(\varepsilon)$ can be better represented on the basis of the mean volume radius, than by using $N_c$ alone. Furthermore, Liu et al. (2008) proposed an new analytical expression that represents $\varepsilon$ in terms of a function of the ratio of the liquid water content to the droplet number concentration $L_c/N_c$ (Liu relationship):

$$\beta(\varepsilon) = 0.07(\frac{L_c}{N_c})^{-0.14}. \tag{6}$$

According to the equation of parameterization of $\beta(\varepsilon)$, $\varepsilon$ can be expressed as the equation in terms of $\beta(\varepsilon)$

$$\varepsilon = [-\frac{1}{2} + \frac{1}{8}\beta(\varepsilon)^3 + \frac{1}{8}\sqrt{8\beta(\varepsilon)^3 + \beta(\varepsilon)^6}]^{\frac{1}{2}}. \tag{7}$$

Note that Rotstayn and Liu (2009) applied both Expression (2) and Expression (6) to the microphysical scheme of a low-resolution version of the CSIRO GCM and discussed their influences on the corresponding model results.

The Morrison-Grabowski relationship is based on small number of measurements ($\varepsilon = 0.33$ for maritime air masses and $\varepsilon = 0.43$ for continental air masses) reported in Martin et al., 1994, while the Rotstayn-Liu relationship is derived from more measurements described by Liu and Daum (2002). Also, the Rotstayn-Liu relationship assumes the dispersion levels off at approximately 800 cm$^{-3}$ while the linear Morrison-Grabowski relationship has no such limit. As a reference, Figure 1 compares the four different relationships between $\varepsilon$ and the cloud droplet number concentration $N_c$ including $\varepsilon$ fixed as 0.4, the

Morrison-Grabowski relationship, the Rotstayn-Liu relationship, and the Liu relationship. The fixed value of $\varepsilon$ ($\varepsilon = 0.4$) denotes the average value based on Zhao et al. (2006). This relationship with fixed $\varepsilon$ does not consider the dispersion effect. The other three relationships all show that $\varepsilon$ is an increasing function of the cloud droplet number concentration with different slopes $\Delta\varepsilon/\Delta N_c$. The Liu relationship ($\varepsilon$-$L_c/N_c$) has the largest slope, especially at low droplet concentrations, followed by the Rotstayn-Liu relationship and Morrison-Grabowski relationship. Note that the slope ($\Delta\varepsilon/\Delta N_c$) for the Liu relationship (Expressions (6) and (7)) is also dependent on the liquid water content $L_c$, decreasing with increasing $L_c$ in Figure 1 as also discussed by Rotstayn and Liu (2009).

## 3 Description and evaluation of CAM5.1

### 3.1 CAM5.1 and set-up of the simulations

The GCM here used in this study is the version 5.1 of the Community Atmosphere Model labeled as CAM5.1 (the atmospheric component of the Community Earth System Model (CESM 1.0.3)), which is documented in Neale et al. (2010). A physically-based treatment of aerosol-cloud-climate interactions in stratiform clouds was implemented to allow for effective investigation of the aerosol direct effect, semi-direct effect, and indirect effect, which are fully described in Ghan et al. (2012) and Ghan (2013). The CAM5.1 includes a 3-mode version of the modal aerosol model (MAM3 scheme), which can simulate internal mixtures of sulfate, organics, black carbon, dust, and sea-salt (Liu et al., 2012). This model includes a detailed treatment of cloud microphysics by linking a two-moment bulk cloud microphysics scheme (Morrison and Gettelman 2008) to the MAM3 scheme with detailed descriptions of ice nucleation and droplet activation of cloud drops (Gettelman et al., 2010; Neale et al., 2010). The longwave and shortwave radiation codes are based upon the Rapid Radiative Transfer Model developed for application to GCMs (RRTMG) as described by Iacono et al. (2008). The parameterizations of $R_e$ and $Au$ are described by Morrison and Gettelman (2008), where we used the equations with $\varepsilon$ of (2) and (3) instead of the existing parameterization in the CAM5.1 model.

The CAM5.1 simulations were conducted with the finite-volume dynamical core with 30 vertical layers from the surface to 2 hPa at a horizontal grid resolution of $1.9° \times 2.5°$. All the simulations were performed for ten years after a one-year spin-up with fixed climatological sea-ice extent and sea surface temperatures, as well as levels of greenhouse gases for the year 2000. The model time step is 30 minutes for all the simulations in this study. Anthropogenic aerosol emissions including black carbon, organics, and sulfate are derived from the IPCC AR5 emission data set (Lamarque et al., 2010) for the year 2000 (PD experiment) and for the year 1850 (PI experiment). Results of the PD experiment are used to compare to the observed data for evaluating the model we used in Subsection 3.2. The difference between the simulations with the same ocean surface conditions but aerosol emissions for PD and PI was used to calculate the changes in cloud microphysical properties and cloud radiative forcing induced by anthropogenic aerosols in Section 4. Note that the AIF is the combined first and second indirect forcing, which is the effective radiative forcing (net TOA radiative fluxes to perturbations with rapid adjustments), not instantaneous radiative forcing, following IPCC (2013).

To examine the influences of different parameterizations of effective radius, autoconversion process, and $\varepsilon$, five numerical experiments (Old, New1, New2, New3, and New4) were performed with the different aerosol emission data including PD and PI. The Old experiment (Old) uses the standard microphysics scheme of CAM5.1 (Morrison and Gettelman, 2008). Compared to Old with the standard microphysics scheme, the four New experiments (News) were conducted by use of the new cloud microphysical parameterizations of $R_e$ (2) and two-moment $Au$ (3) with four different ways of representing $\varepsilon$ including fixed $\varepsilon = 0.4$ (New1), the Morrison-Grabowski relationship (New2), the Rotstayn-Liu relationship (New3), and the Liu relationship (New4). Note that the New1 experiment with $\varepsilon$ fixed at 0.4 does not account for the dispersion effect, whereas the other three experiments (New2, New3, and New4) consider the dispersion effect differently, permitting systematic evaluation of the importance of representing anthropogenic aerosols on $\varepsilon$ in determining AIF and other key cloud and precipitation properties. For convenience, key characteristics of the five simulations with the two different aerosol emission data are summarized in Table 1.

## 3.2 Evaluation of the influences of the new parameterizations

### 3.2.1 Annual global means

Table 2 summarized the key properties derived from the five different model experiments in PD and the corresponding observational data including aerosol optical depth at wavelength 550 nm (AOD), liquid water path (LWP, g m$^{-2}$), ice water path (IWP, g m$^{-2}$), vertical integrated cloud droplet number concentration (N$_d$, $10^{10}$ m$^{-2}$), cloud top effective radius (REL, $\mu$m), total cloud fraction (CLDTOT, %), low cloud fraction (CLDLOW, %), middle cloud fraction (CLDMED, %), high cloud fraction (CLDHGH,%), total precipitation rate (P$_{tot}$, mm day$^{-1}$), shortwave cloud radiative forcing (SWCF, W m$^{-2}$), and longwave cloud radiative forcing (LWCF, W m$^{-2}$).

The values of AOD derived from the five simulations are similar, ranging from 0.121 to 0.125. Because the same anthropogenic emissions are used in all the simulations, the small differences between the simulated AODs are likely due to the differences in the meteorological parameters that can influence the aerosol emission (especially the natural aerosols, e. g., mineral dust and sea salt), transport, and lifetime of aerosols and thus AOD. All the simulated values of AOD are much smaller than that (0.15) derived from the satellite retrieval composite by Kinne et al. (2006), suggesting that CAM5.1 underestimates AOD as compared to satellite observations. It is shown that anthropogenic aerosol emissions are underestimated, especially in East and South Asia, which leads to the low bias of the CAM5.1 simulated AOD in comparison with the observational data including the AERONET and satellite data (Liu et al., 2012).

The LWP produced by all simulations approximately falls within the range from 36 to 45 g m$^{-2}$. The simulated LWP is lower in News (including New1, New2, New3, and New4) than that in Old. The incorporation of the new autoconversion parameterization in CAM5.1 has more efficient autoconversion process to form rain drops and leads to a decrease in the LWP, primarily because that this new cloud parameterization can yield a larger autoconversion rate compared to the KK parameterization used in the standard CAM5.1 (Wood, 2005). It is also noted that there is a significant difference in LWP between the four New experiments because the different parameterizations of $\varepsilon$ will affect the autoconversion rate by equation

(3), and thereby change the simulated LWP. The behavior of IWP is opposite to LWP, with IWP being larger in News than that in Old. Compared to the differences in LWP between the four New experiments, the differences in the corresponding IWPs are less noticeable. Note that all the GCMs including CAM5.1 distinguish between smaller cloud droplets and larger rain drops artificially, the simulated LWP is directly related to cloud droplets. However, the observed LWP by satellite retrievals is the sum of the cloud water path and the rain water path, and additionally it cannot be retrieved reliably (Lohmann et al., 2007; Posselt and Lohmann, 2008; Gettelman et al., 2015). The method of the observed IWP by satellite retrievals is similar with that of the observed LWP by satellite retrievals. Therefore, the observational values for LWP and LWP from satellite retrievals are not reported in the table.

The column cloud droplet number concentration $N_d$ derived from all CAM5.1 simulations ranges from $1.33 \times 10^{10}$ to $1.47 \times 10^{10}$ m$^{-2}$, all of which are markedly lower than that ($4.01 \times 10^{10}$ m$^{-2}$) derived from the Advanced Very High Resolution Radiometer (AVHRR) retrieval (Han et al., 1998). Hence, CAM5.1 severely underestimates the column cloud droplet number concentration. The global annual average value of effective radius (REL) is 9.21 $\mu m$ in Old, which shows an underestimation of REL in the satellite observation. Compared to REL in Old, the simulated REL in News (from 10.08 to 11.48 $\mu m$) becomes much larger, which is in better agreement with the satellite observational value of 10.5 $\mu m$ (Han et al., 1998). It is noted that the simulated cloud droplet number concentration is underestimated in CAM5.1 model while the effective radius agrees with satellites. This apparent inconsistency suggests that the simulated liquid water content may be somehow underestimated. Unfortunately, we do not have observed cloud water content to verify this (Gettelman et al., 2015). The simulated total cloud cover (65.50%, 65.63%, 65.74%, and 65.82%) in News are larger than that (64.02%) in Old, and in better agreement with the observational range of 65−75% obtained from the MODIS, ISCCP and HIRS data (Rossow and Schiffer, 1999; King et al., 2003; Wylie et al., 2005). The low cloud fraction, middle cloud fraction and high cloud fraction are also increased in News compared to that in Old.

The simulated total precipitation rate in Old is about 2.96 mm day$^{-1}$, and it is the same as 2.97 mm day$^{-1}$ in the four New experiments, which are all larger than that (2.67 mm day$^{-1}$) in GPCP observations for the years 1979−2009 (Alder et al., 2003). Hence the global annual mean precipitation is overestimated in all the CAM5.1 simulations. The SWCF and LWCF of satellite observations are from the CERES-EBAF estimates for the years 2000−2010 from Loeb et al. (2009). The simulated values of SWCF with the range from −49.82 to −53.03 W m$^{-2}$ are overestimated in Old and News, as compared to the value of −47.07 W m$^{-2}$ in observations, whereas the LWCF in all CAM5.1 simulations (from 24.06 to 25.51 W m$^{-2}$) is underestimated compared to the observational value 26.48 W m$^{-2}$ from CERES-EBAF estimates.

### 3.2.2   Annual and seasonal, zonal means

To further explore differences between the various cloud microphysical parameterizations, we use physical variables derived from observations including SWCF, LWCF, and surface precipitation to make a detailed comparison for annual and seasonal zonal means, because these three physical variables are very important and all from more reliable field observations. Annual, JJA (June, July and August) and DJF (December, January and February) zonal means of SWCF in all CAM5.1 simulations and CERES-EBAF observations, as well as their corresponding differences between models and observations are shown in Figure

2. The zonal mean tendencies of SWCF in all CAM5.1 simulations are in better agreement with CERES-EBAF retrievals for annual and seasonal zonal means. All the simulated SWCF is much overestimated as compared to the CERES-EBAF observations in low-latitudes for Annual, JJA and DJF means (Figures 2a, 2c, and 2e). Compared to Old, the corresponding simulated SWCF in News is reduced effectively and much closer to the observations over low-latitudes regions, as seen from Figures 2b, 2d, and 2f. The autoconversion rate used here is larger than the autoconversion rate of CAM5.1, especially at larger cloud water, which leads to less liquid clouds and smaller SWCF over low latitude regions. No significant differences in the spatial pattern correlation coefficients are found between the Old and the four New experiments. However, in Annual, JJA and DJF means, the root-mean-square error (RMSE) in comparison with observations are all reduced significantly in News, with respect to that in Old. These results indicate that, the new cloud parameterizations that explicitly account for dispersion effect better simulate the shortwave cloud radiative forcing for annual and seasonal, zonal means, especially in terms of RMSE.

Figure 3 shows the annual, JJA and DJF zonal means of LWCF in all CAM5.1 simulations and CERES-EBAF observations, as well as their corresponding differences between models and observations. The simulated LWCFs by all simulations are nearly the same as the simulated SWCF, which are also in better agreement with CERES retrievals for annual and seasonal zonal means. Evidently, the LWCF in all the simulations is overestimated in low-latitudes, whereas it is underestimated in middle and high-latitudes (Figure 3a, 3c, and 3e). The simulated LWCF in News is much larger over low-latitude regions compared to Old (Figures 3b, 3d, and 3f). However, the corresponding simulated LWCF in News is increased significantly over every latitude, which is much closer to the CERES-EBAF observations over middle and high-latitudes. That is because of larger higher cloud fraction in NEWs compared to that in Old (Table 2). It can be further seen that, the annual and seasonal global mean values in News are all closer to the CERES-EBAF observations compared to Old from Table 4. The New experiments also exhibit a slightly higher spatial pattern correlation coefficient compared to Old. The influences on the RSME of annual, JJA, and DJF means are minimal between Old and News.

Figure 4 shows annual and seasonal zonal means of total precipitation rate in all simulations and GPCP observations, as well as their corresponding differences between models and observations. The simulated precipitation rate is overestimated in low-latitudes, while it is underestimated in middle and high-latitudes as shown in Figures 4a, 4c, and 4e. It is further shown that the simulated precipitation in News is slightly changed in comparison with that in Old for the annual and seasonal zonal (Figures 4b, 4d, and 4f) and global means (Table 5). The RSME of annual, JJA, and DJF mean in comparison with observations is slightly reduced in News, and the spatial pattern correlation coefficients is also slightly increased from Old to News in Table 5. This is because that all the CAM5.1 simulations were conducted with the same sea surface temperature and the same ice content, governing the rate of water evaporation processes from the sea surface. The equilibrium of amount in precipitation processes and water evaporation is not affected in any of the simulations, as discussed by Michibata and Takemura (2015). Hence, the CAM5.1 model shows that the differences in surface precipitation are insensitive to various cloud microphysics schemes.

## 4 Dispersion effect on AIF

As discussed in Section 1, consideration of dispersion effect is expected to reduce the first and second aerosol indirect radiative forcings by affecting both the cloud droplet effective radius ($R_e$) and the cloud-to-rain autoconversion process ($Au$) (Liu and Daum, 2002; Liu and Daum, 2004; Xie and Liu, 2009). This section analyzes results of the CAM5.1 simulations to examine the dispersion effects on $R_e$ and $Au$, respectively, and then reevaluate the AIF with the dispersion effects.

### 4.1 Dispersion effect on $R_e$

According to the parameterization of $R_e$ (2) with the different $\varepsilon$-$N_c$ or $\varepsilon$-$L_c/N_c$ relationships, it depicts the variation of $R_e$ with $N_c$, which shows a decreasing $R_e$ with increasing $N_c$ at fixed cloud water content $L_c$ in Figure 5. The dependence of $R_e$ on $N_c$ illustrates the first aerosol indirect effect, leading to enhanced cloud albedo and cloud radiative radiative forcing. In comparison with the fixed dispersion (0.4), the other $\varepsilon$-$N_c$ or $\varepsilon$-$L_c/N_c$ relationships with dispersion effect can reduce the magnitude of variation of $R_e$ effectively, especially for the Rotstayn-Liu and Liu relationships.

Figure 6 presents the annual zonal mean differences in the cloud top effective radius REL ($\Delta$REL) between PD and PI in the four New experiments. It is shown that, compared to $\Delta$REL derived from New1, the $\Delta$REL induced by anthropogenic aerosols can be effectively reduced by dispersion effect from New2, New3, and New4, especially in Northern Hemisphere. The $\Delta$REL for global means (for Northern Hemisphere means) are reduced from $-0.74\ \mu m$ ($-1.24\ \mu m$) in New1 to the range form $-0.38$ $\mu m$ to $-0.67\ \mu m$ (from $-0.63\ \mu m$ to $-1.13\ \mu m$) in New2, New3, and New4 with dispersion effect in Table 6. Based upon the physical principle for dispersion effect (as seen from Figure 1), the cloud droplet number concentration induced by more anthropogenic aerosols from anthropogenic activities are remarkably increased in Northern Hemisphere, which shows a larger increase in $\varepsilon$, and then a larger reduction in $\Delta$REL, compared to the Southern Hemisphere. Hence, dispersion effect is stronger over the Northern Hemisphere than over the Southern Hemisphere (Liu et al., 2008). Therefore, the increase of $\Delta$REL with dispersion effect leads to a warming effect and offsets the cooling from the increased droplet concentration alone, especially in Northern Hemisphere.

From Table 7 in terms of differences between New2, New3 and New4, the magnitude of reduction in $\Delta$REL is different compared to New1. The Liu relationship presents a largest magnitude of reduction in $\Delta$REL, and the Rotstayn-Liu relationship is second, and the Morrison-Grabowski relationship gives a smallest magnitude in the global and two hemisphere means, because of different slopes $\Delta\varepsilon/\Delta N_c$ for these $\varepsilon$-$N_c$ or $\varepsilon$-$L_c/N_c$ relationships. The different magnitudes of reduction in $\Delta$REL for these parameterizations of $\varepsilon$ will affect the aerosol first indirect forcing with dispersion effect (Rotstayn and Liu, 2009).

### 4.2 Dispersion effect on $Au$

Based on the parameterization of $Au$ (3) with the different $\varepsilon$-$N_c$ or the $\varepsilon$-$L_c/N_c$ relationships, Figure 7 shows a decreasing $P_L$ ($Au$ in mass content) with increasing $N_c$ at fixed cloud water content $L_c$. The decrease of $P_L$ with increasing $N_c$ shows that the higher cloud droplet concentration leads to a lower autoconversion rate for a given liquid water content, enhancing the cloud lifetime and cloud radiative forcing. Similar with the dispersion effect on $R_e$, the $\varepsilon$-$N_c$ or $\varepsilon$-$L_c/N_c$ relationships

with dispersion effect can also reduce the magnitude of variation of $P_L$ in comparison with $\varepsilon$ fixed as 0.4, where the reducing magnitudes of $P_L$ are also dependent on the parameterizations of $\varepsilon$.

Figure 8 presents the annual zonal mean differences in the liquid water path LWP ($\Delta$LWP) between PD and PI derived from New1, New2, New3, and New4. Compared to $\Delta$LWP in New1, the increased LWP induced by anthropogenic aerosols can be reduced with dispersion effect in New2, New3, and New4, especially in the Northern Hemisphere. These results can also be seen from Table 6. The $\Delta$LWP for global means (for Northern Hemisphere means) can be reduced from 2.01 g m$^{-2}$ (3.10 g m$^{-2}$) in New1 to the range form 1.46 to 1.74 g m$^{-2}$(from 2.16 g to 2.48 g m$^{-2}$) in New2, New3, and New4 with dispersion effect. Nevertheless, the $\Delta$LWP are not always reduced in New2, New3 and New4 because of weaker dispersion effect over the Southern Hemisphere. Hence, the reduction of $\Delta$LWP with dispersion effect can exert a warming effect and offset the cooling from the convetional second aerosol indirect effect that considers only the influence from the increased droplet concentration alone. It is also shown that the magnitude of reduction in $\Delta$LWP in New2, New3 and New4 is different compared to New1 from Table 7, which is dependent on the different slopes $\Delta\varepsilon/\Delta N_c$ for the different parameterizations of $\varepsilon$.

It is noted that different parameterizations of the autoconversion process have been coupled to GCMs, showing that the $\Delta$LWP induced by aerosols can be significantly changed by them and will affect the aerosol second indirect effects (Penner et al., 2006; Chuang et al., 2012), which is consistent with our results. Additionally, Guo et al. (2008) also pointed that the threshold functions associated with the autoconversion process can significantly influence the cloud fraction, and the liquid water path, and therefore affect the second aerosol indirect forcing. Hence, various threshold functions maybe influence the corresponding change of cloud microphysical and radiative properties induced by increased aerosols by affecting autoconversion processes.

## 4.3 Evaluation of AIF including dispersion effect

This subsection evaluates the aerosol indirect forcing (AIF), which can be defined as the changes in total cloud radiative effect including the shortwave and longwave cloud radiative forcing with and without anthropogenic aerosols. Table 6 shows the global, Northern Hemisphere and Southern Hemisphere annual mean changes of liquid water path ($\triangle$LWP), cloud top effective radius ($\triangle$REL), shortwave cloud radiative forcing ($\triangle$SWCF), longwave cloud radiative forcing ($\triangle$LWCF) and total cloud radiative forcing (AIF) induced by aerosols in News. With an increase in anthropogenic aerosols, the LWP can be increased by the decreased autoconversion rate to form rain drops, and additionally the REL can be reduced significantly due to the enhanced activation of aerosols to cloud droplets (Xie et al., 2013). Due to the increased LWP and the decreased REL, the SWCF and LWCF can be increased by anthropogenic aerosols, and the total cloud radiative forcing (SWCF+LWCF) can also be increased, where the aerosol-induced SWCF is dominated for changes in the total cloud radiative forcing. Because of higher AOD induced by anthropogenic aerosols over Northern Hemisphere (Ghan et al., 2013), $\triangle$LWP and $\triangle$REL are larger (Figure 6 and Figure 8), leading to larger $\triangle$SWCF, $\triangle$LWCF and AIF than that over Southern Hemisphere (Figure 9). These results are very similar between these four New experiments, which are consistent with some previous studies (as reviewed by Lohmann, et al., 2005).

Figure 8 shows the differences in the aerosol-induced SWCF, LWCF, and AIF between these four New cloud microphysical parameterizations (New1, New2, New3, and New4). Considering dispersion effect on the reduction in $\Delta$REL and $\Delta$LWP, the aerosol-induced SWCF and AIF are significantly decreased in New2, New3, and New4 in comparison with New1, especially in the Northern Hemisphere. The aerosol-induced change in LWCF is insignificant compared to the corresponding SWCF and AIF. The difference between the two hemispheres shows that dispersion effect over the Northern Hemisphere is much stronger than that over the Southern Hemisphere, compensating the hemispheric contrasts induced by their difference in droplet concentration (Liu et al., 2008). As also shown in Table 7, the changes in annual global mean SWCF are significantly decreased by 0.18 W m$^{-2}$ (New2), 0.23 W m$^{-2}$ (New3), 0.26 W m$^{-2}$ (New4) in comparison with New1. The changes in annual global mean LWCF are slightly decreased by $-0.09$ W m$^{-2}$(New2), $-0.02$ W m$^{-2}$ (New3), and $-0.10$ W m$^{-2}$ (New4). In comparison with New1, the AIF are decreased by 0.10 W m$^{-2}$ (New2), 0.21 W m$^{-2}$ (New3), and 0.16 W m$^{-2}$ (New4) for global scale, as well as by a bigger margin from 0.25 and 0.39 W m$^{-2}$ for Northern Hemisphere, because of stronger dispersion effect over this Hemisphere. Note that, the three $\varepsilon$-$N_c$ or $\varepsilon$-$L_c/N_c$ relationships show different magnitudes of reduction in aerosol-induced SWCF, as well as AIF, due to different $\Delta\varepsilon/\Delta N_c$ as shown in Figure 1. As expected, the Liu relationship with $\varepsilon$-$L_c/N_c$ presents a largest magnitude of reduction in the aerosol-induced SWCF because of the largest $\Delta\varepsilon/\Delta N_c$ compared to the fixed $\varepsilon$, the second one is the Rotstayn-Liu relationship with $\varepsilon$-$N_c$, the smallest one is the the Morrison-Grabowski relationship with $\varepsilon$-$N_c$. These results are similar with the results of Rotstayn and Liu (2009). Nevertheless, the magnitudes of reduction in AIF are changed for these relationships when considering the aerosol-induced LWCF. Note that, for the Rotstayn-Liu and Liu relationships, they can also yield a stronger dispersion effect on AIF compared to the Morrison-Grabowski relationship.

In general, due to dispersion effects on $R_e$ and $Au$, the changes induced by anthropogenic aerosols in the cloud droplet effective radius and the liquid water path are decreased significantly, and the AIF are also reduced by a range of 0.10 to 0.21 W m$^{-2}$ for global scale, and by a bigger margin (from 0.25 to 0.39 W m$^{-2}$) for the Northern Hemisphere for the two $\varepsilon$-$N_c$ and the $\varepsilon$-$L_c/N_c$ relationships in comparison with that in fixed $\varepsilon$ with 0.4, because of stronger dispersion effect over this hemisphere. The magnitude of reduction in AIF with dispersion effect are mainly dependent on the slopes $\Delta\varepsilon/\Delta N_c$ for the two $\varepsilon$-$N_c$ and the $\varepsilon$-$L_c/N_c$ relationships. It is worth noting that the reduction of AIF induced by dispersion effect in this study is much smaller than that (approximately $-0.5$ W m$^{-2}$ for global means) reported by Rotstayn and Liu (2005). This difference lies likely in the reference autocnversion parameterations. In this study, Eq. (3) with fixed dispersion of 0.4 is used whereas Rotstayn and Liu (2005) used a different one give in $P_L = E_c\pi\kappa_1(\frac{3}{4\pi\rho_l})N_c^{-1/3}L_c^{7/3}H(R_3 - R_{3c})$. Hence, we believe that the difference in Rotstayn and Liu (2005) includes not only the dispersion effect but also different autoconversion parameterizations whereas our results just represent the dispersion effect. Additionally, here we used the complete two-moment autoconversion parameterizations with relative dispersion including droplet number concentration and mass content, and Rotstayn and Liu (2005) only adopted the mass content autoconversion parameterization (Liu and Daum, 2004), which also results in the differences of the reduced AIF .

## 5 Concluding Remarks

In order to accurately evaluate the dispersion effect with GCMs, especially on AIF, we first implement the complete cloud microphysical parameterizations of $R_e$ and the two-moment $Au$ with $\varepsilon$ into CAM 5.1 in this study. We then perform and analyze a suite of sensitivity experiments of $\varepsilon$ with fixed value as 0.4, the two positive $\varepsilon$-$N_c$ relationships (the Morrison-Grabowski and the Rotstayn-Liu relationships), and the $\varepsilon$-$L_c/N_c$ relationship (the Liu relationship). These results show that the parameterizations that explicitly account for dispersion effect yield a shortwave cloud radiative forcing is much better compared to the standard model. Consideration of dispersion effect can significantly decrease the aerosol-induced changes in the cloud top effective radius and the liquid water path, especially in Northern Hemisphere. The corresponding AIF with dispersion effect is also reduced remarkably by a range from 0.10 to 0.21 W m$^{-2}$ for global scale, and by a bigger margin from 0.25 to 0.39 W m$^{-2}$ for the Northern Hemisphere for these two different $\varepsilon$-$N_c$, and the $\varepsilon$-$L_c/N_c$ relationships in comparison with that in fixed $\varepsilon$ with 0.4, where the magnitudes of reduction in AIF are mainly dependent on the slopes $\Delta\varepsilon/\Delta N_c$ of the parameterizations of $\varepsilon$.

It is noted that, compared to the $\varepsilon$-$N_c$ relationships (the Morrison-Grabowski and the Rotstayn-Liu relationships), the new parameterization of $\varepsilon$ in terms of $L_c/N_c$ (the Liu relationship) can also account for the effect of variations in $L_c$, showing a larger $\Delta\varepsilon/\Delta N_c$ at low $L_c$ as shown in Figure 1. Hence, the Liu relationship can yield a much stronger dispersion effect in terms of AIF over polluted/continental regions with low $L_c$, compared to these $\varepsilon$-$N_c$ relationships (Rotstayn and Liu, 2009). Hence, the spatial difference (e.g., Land vs. Ocean or inland vs. coastal regions) of dispersion effect in AIF between the Liu relationship and other $\varepsilon$-$N_c$ relationships derived from CAM5.1 will be analysised in the future. Additionally, as discussed above, the threshold functions associated with the autoconversion process can significantly influence the macrophysical and microphysical properties, as well as the second aerosol indirect forcing (Guo et al., 2008).

Our systematic investigation of dispersion effect through both effective radius and autoconversion rate with CAM5.1 reinforces previous studies on the importance of considering the dispersion effect in climate models (Peng and Lohmann, 2003; Rotstayn and Liu, 2003; Rotstayn and Liu, 2005; Rotstayn and Liu 2009). It is noted that the factors including the aerosol chemical, physical and atmosphere environmental factors determining $\varepsilon$ and the relationships to cloud droplet number concentration $N_c$ or other cloud microphysical properties (e.g., water per droplet $L_c/N_c$) remain poorly understood (Zhao et al., 2006; Peng et al., 2007; Liu et al., 2008). Hence, in-depth explorations of the relationships between $\varepsilon$ and cloud microphysical properties are needed to further improve understanding and calculation of the first and second aerosol indirect forcings.

*Acknowledgements.* The authors thank the two anonymous reviewers for valuable comments and suggestions. This work was jointly supported by National Key Research and Development Program of China (2016YFA0601904) and the National Natural Science Foundation of China (41690115, 41572150). H. Zhang is supported by the National Natural Science Foundation of China (61432018). Yiran Peng is supported by 973 project 2014CB441302. Y. Liu is supported by the US Department of Energy's Atmospheric System Research (ASR) program.

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

**Table 1.** Description of simulations performed in our study.

| Simulation | Paremeterization | Simulated time | aerosol emissions (PD) | aerosol emissions (PI) |
|:---:|:---:|:---:|:---:|:---:|
| Old | Standard scheme of CAM5.1 | 10 years | AR5 2000 | AR5 1850 |
| New1 | Fixed $\varepsilon$ ($\varepsilon$=0.4) | 10 years | AR5 2000 | AR5 1850 |
| New2 | Morrison-Grabowski relationship | 10 years | AR5 2000 | AR5 1850 |
| New3 | Rotstayn-Liu relationship | 10 years | AR5 2000 | AR5 1850 |
| New4 | Liu relationship | 10 years | AR5 2000 | AR5 1850 |

**Table 2.** Annual global mean aerosols, cloud properties, and surface precipitation, as well as TOA energy budget with year 2000 aerosol emissions including aerosol optical depth at wavelength 550 nm (AOD), liquid water path (LWP), Ice water path (IWP), the vertical integrated cloud droplet number concentration ($N_d$), cloud top effective radius (REL), total cloud fraction (CLDTOT), low cloud fraction (CLDLOW), middle cloud fraction (CLDMED), high cloud fraction (CLDHGH), total precipitation rate ($P_{tot}$), shortwave cloud radiative forcing (SWCF), and longwave cloud radiative forcing (LWCF).

| Simulation | Old | New1 | New2 | New3 | New4 | OBS |
|---|---|---|---|---|---|---|
| AOD | 0.121 | 0.122 | 0.122 | 0.124 | 0.125 | $0.15^a$ |
| LWP, g m$^{-2}$ | 44.74 | 36.76 | 40.33 | 37.62 | 43.48 | − |
| IWP, g m$^{-2}$ | 17.78 | 18.70 | 18.88 | 18.84 | 18.96 | − |
| $N_d$, $10^{10}$ m$^{-2}$ | 1.38 | 1.33 | 1.40 | 1.35 | 1.47 | $4.01^b$ |
| REL, $\mu$m | 9.21 | 11.48 | 10.87 | 11.32 | 10.08 | $10.5^b$ |
| CLDTOT, % | 64.02 | 65.50 | 65.63 | 65.74 | 65.82 | $65-75^c$ |
| CLDLOW, % | 43.61 | 44.88 | 45.25 | 45.31 | 45.47 | − |
| CLDMID, % | 27.27 | 27.58 | 27.67 | 27.65 | 27.72 | − |
| CLDHGH, % | 38.09 | 39.24 | 39.09 | 39.22 | 39.16 | $21-33^d$ |
| $P_{tot}$, mm day$^{-1}$ | 2.96 | 2.97 | 2.97 | 2.97 | 2.97 | $2.67^e$ |
| SWCF, W m$^{-2}$ | −52.08 | −49.82 | −52.40 | −51.01 | −53.03 | $-47.07^f$ |
| LWCF, W m$^{-2}$ | 24.06 | 25.23 | 25.40 | 25.37 | 25.51 | $26.48^f$ |

[a] AOD is from a satellite retrieval composite (Kinne et al., 2006), [b] $N_d$ and REL are from the AVHRR data (Han et al., 1998). [c] CLDTOT is obtained from ISCCP (Rossow and Schiffer, 1999), MODIS data (Platnick et al., 2003), and HIRS data (Wylie et al., 2005). [d] CLDHGH is obtained from ISCCP data (Rossow and Schiffer, 1999) and HIRS data (Wylie et al., 2005). [e] $P_{tot}$ is taken from the Global Precipitation Climatology Project (GPCP) for the years 1979−2009 (Adler et al., 2003). [f] Radiative fluxes from the CERES-EBAF are for the years 2000−2010 from Loeb et al. (2009).

**Table 3.** Mean (annual and seasonal global mean values) and Model-OBS (the difference of annual and seasonal global mean values between models and observations), RMSE (root mean squared error), and R (spatial pattern correlation) of the modeling results compared to the observed SWCF from CERES-EBAF for ANN, JJA, DJF.

|  |  | ANN | JJA | DJF |
|---|---|---|---|---|
| OBS (W m$^{-2}$) | Mean | $-47.07$ | $-44.36$ | $-51.65$ |
| Old (W m$^{-2}$) | Mean | $-52.08$ | $-52.98$ | $-54.01$ |
|  | Model$-$OBS | $-5.01$ | $-8.62$ | $-2.36$ |
|  | RMSE(R) | 16.50(0.77) | 22.03(0.84) | 22.24(0.82) |
| New1 (W m$^{-2}$) | Mean | $-49.82$ | $-50.47$ | $-51.63$ |
|  | Model$-$OBS | $-2.75$ | $-6.11$ | 0.02 |
|  | RMSE(R) | 15.84(0.76) | 20.58(0.84) | 21.84(0.82) |
| New2 (W m$^{-2}$) | Mean | $-52.40$ | $-53.21$ | $-54.45$ |
|  | Model$-$OBS | $-5.33$ | $-8.85$ | $-2.80$ |
|  | RMSE(R) | 16.28(0.78) | 21.68(0.85) | 21.60(0.83) |
| New3 (W m$^{-2}$) | Mean | $-51.01$ | $-51.49$ | $-53.01$ |
|  | Model$-$OBS | $-3.94$ | $-7.14$ | $-1.37$ |
|  | RMSE(R) | 15.74(0.77) | 20.69(0.84) | 21.62(0.83) |
| New4 (W m$^{-2}$) | Mean | $-53.04$ | $-53.90$ | $-54.98$ |
|  | Model$-$OBS | $-5.96$ | $-9.54$ | $-3.34$ |
|  | RMSE(R) | 16.42(0.78) | 21.80(0.85) | 21.67(0.83) |

**Table 4.** Mean (annual and seasonal global mean values) and Model-OBS (the difference of annual and seasonal global mean values between models and observations), RMSE (root mean squared error), and R (spatial pattern correlation) of the modeling results compared to the observed LWCF from CERES-EBAF for ANN, JJA, DJF.

| | | ANN | JJA | DJF |
|---|---|---|---|---|
| OBS (W m$^{-2}$) | Mean | 26.48 | 26.60 | 26.16 |
| Old (W m$^{-2}$) | Mean | 24.06 | 24.74 | 23.10 |
| | Model−OBS | −2.42 | −1.86 | −3.06 |
| | RMSE(R) | 7.13(0.87) | 10.42(0.83) | 9.06(0.88) |
| New1 (W m$^{-2}$) | Mean | 25.24 | 25.92 | 24.34 |
| | Model−OBS | −1.24 | −0.68 | −1.82 |
| | RMSE(R) | 7.20(0.88) | 10.60(0.84) | 9.19(0.90) |
| New2 (W m$^{-2}$) | Mean | 25.41 | 26.14 | 24.44 |
| | Model−OBS | −1.07 | −0.46 | −1.72 |
| | RMSE(R) | 7.03(0.88) | 10.53(0.84) | 9.20(0.89) |
| New3 (W m$^{-2}$) | Mean | 25.37 | 26.04 | 24.41 |
| | Model−OBS | −1.11 | −0.56 | −1.75 |
| | RMSE | 7.12(0.88) | 10.45(0.85) | 9.38(0.89) |
| New4 (W m$^{-2}$) | Mean | 25.52 | 26.27 | 24.47 |
| | Model−OBS | −0.96 | −33 | −1.69 |
| | RMSE(R) | 6.96(0.88) | 10.42(0.84) | 9.01(0.90) |

**Table 5.** Mean (annual and seasonal global mean values) and Model-OBS (the difference of annual and seasonal global mean values between models and observations), RMSE (root mean squared error), and R (spatial pattern correlation) of the modeling results compared to the observed precipitation rate from GPCP for ANN, JJA, DJF.

| | | ANN | JJA | DJF |
|---|---|---|---|---|
| OBS (mm day$^{-1}$) | Mean | 2.67 | 2.70 | 2.67 |
| Old (mm day$^{-1}$) | Mean | 2.96 | 3.04 | 2.95 |
| | Model−OBS | 0.29 | 0.34 | 0.28 |
| | RMSE(R) | 1.09(0.86) | 1.67(0.81) | 1.41(0.85) |
| New1 (mm day$^{-1}$) | Mean | 2.97 | 3.05 | 2.96 |
| | Model−OBS | 0.30 | 0.35 | 0.29 |
| | RMSE(R) | 1.06(0.87) | 1.64(0.82) | 1.37(0.86) |
| New2 (mm day$^{-1}$) | Mean | 2.97 | 3.04 | 2.96 |
| | Model−OBS | 0.30 | 0.34 | 0.29 |
| | RMSE(R) | 1.06(0.87) | 1.62(0.83) | 1.39(0.86) |
| New3 (mm day$^{-1}$) | Mean | 2.97 | 3.06 | 2.95 |
| | Model−OBS | 0.30 | 0.35 | 0.28 |
| | RMSE(R) | 1.06(0.87) | 1.62(0.83) | 1.40(0.86) |
| New4 (mm day$^{-1}$) | Mean | 2.97 | 3.05 | 2.96 |
| | Model−OBS | 0.30 | 0.35 | 0.29 |
| | RMSE(R) | 1.07(0.87) | 1.63(0.82) | 1.36(0.87) |

**Table 6.** Global, Northern Hemisphere (NH) and Southern Hemisphere (SH) annual mean changes of cloud top effective radius ($\triangle$REL), liquid water path ($\triangle$LWP), shortwave cloud radiative forcing ($\triangle$SWCF), longwave cloud radiative forcing ($\triangle$LWCF) between PD and PI, as well as aerosol indirect forcing (AIF, W m$^{-2}$) in News.

| | | $\triangle$ REL ($um$) | $\triangle$LWP (g m$^{-2}$) | $\triangle$ SWCF | $\triangle$LWCF | AIF (W m$^{-2}$) |
|---|---|---|---|---|---|---|
| New1 | Global | −0.74 | 2.01 | −2.13 | 0.64 | −1.49 |
| | NH | −1.24 | 3.10 | −3.15 | 1.06 | −2.09 |
| | SH | −0.24 | 0.91 | −1.12 | 0.23 | −0.89 |
| New2 | Global | −0.67 | 1.74 | −1.95 | 0.55 | −1.39 |
| | NH | −1.13 | 2.48 | −2.68 | 0.84 | −1.84 |
| | SH | −0.21 | 0.99 | −1.22 | 0.27 | −0.95 |
| New3 | Global | −0.65 | 1.46 | −1.90 | 0.62 | −1.28 |
| | NH | −1.10 | 2.35 | −2.74 | 0.95 | −1.79 |
| | SH | −0.20 | 0.57 | −1.06 | 0.29 | −0.77 |
| New4 | Global | −0.38 | 1.67 | −1.87 | 0.54 | −1.33 |
| | NH | −0.63 | 2.16 | −2.39 | 0.68 | −1.70 |
| | SH | −0.12 | 1.18 | −1.35 | 0.39 | −0.96 |

**Table 7.** Differences (New2-New1, and New3-New1, and New4-New1) in global, Northern hemisphere (NH) and Southern hemisphere (SH) annual mean changes of cloud top effective radius ($\triangle$REL), liquid water path ($\triangle$LWP), shortwave cloud radiative forcing ($\triangle$SWCF), longwave cloud radiative forcing ($\triangle$LWCF) between PD and PI, as well as aerosol indirect forcing (AIF).

|           |        | $\triangle$REL ($um$) | $\triangle$LWP (g m$^{-2}$) | $\triangle$SWCF | $\triangle$LWCF | AIF (W m$^{-2}$) |
|-----------|--------|----------|----------|----------|----------|----------|
| New2-New1 | Global | 0.07 | $-0.27$ | 0.18 | $-0.09$ | 0.10 |
|           | NH     | 0.11 | $-0.62$ | 0.47 | $-0.22$ | 0.25 |
|           | SH     | 0.03 | 0.08 | $-0.10$ | 0.04 | $-0.06$ |
| New3-New1 | Global | 0.09 | $-0.55$ | 0.23 | $-0.02$ | 0.21 |
|           | NH     | 0.14 | $-0.75$ | 0.41 | $-0.11$ | 0.30 |
|           | SH     | 0.04 | $-0.34$ | 0.06 | 0.06 | 0.12 |
| New4-New1 | Global | 0.36 | $-0.34$ | 0.26 | $-0.10$ | 0.16 |
|           | NH     | 0.61 | $-0.94$ | 0.76 | $-0.38$ | 0.39 |
|           | SH     | 0.12 | 0.27 | $-0.23$ | 0.16 | $-0.07$ |

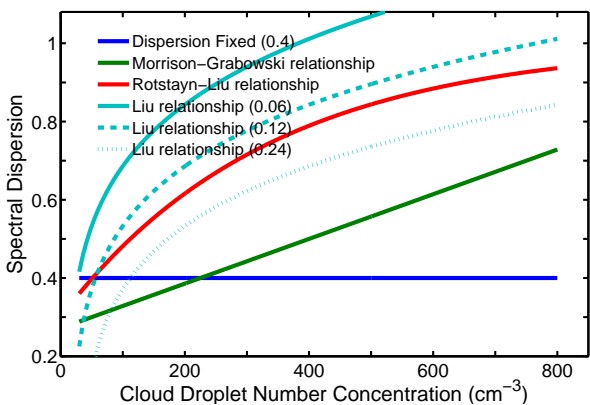

**Figure 1.** Variations in the relative dispersion $\varepsilon$ as the functions of droplet concentration for $\varepsilon$ fixed at 0.4 (blue curve), the Morrison-Grabowski relationship (green curve), the Rotstayn-Liu relationship (red curve), and the Liu relationship with different liquid water content $L_c$ (fixed as 0.06, 0.12 and 0.24 g m$^{-3}$ for different styles of cyan curves).

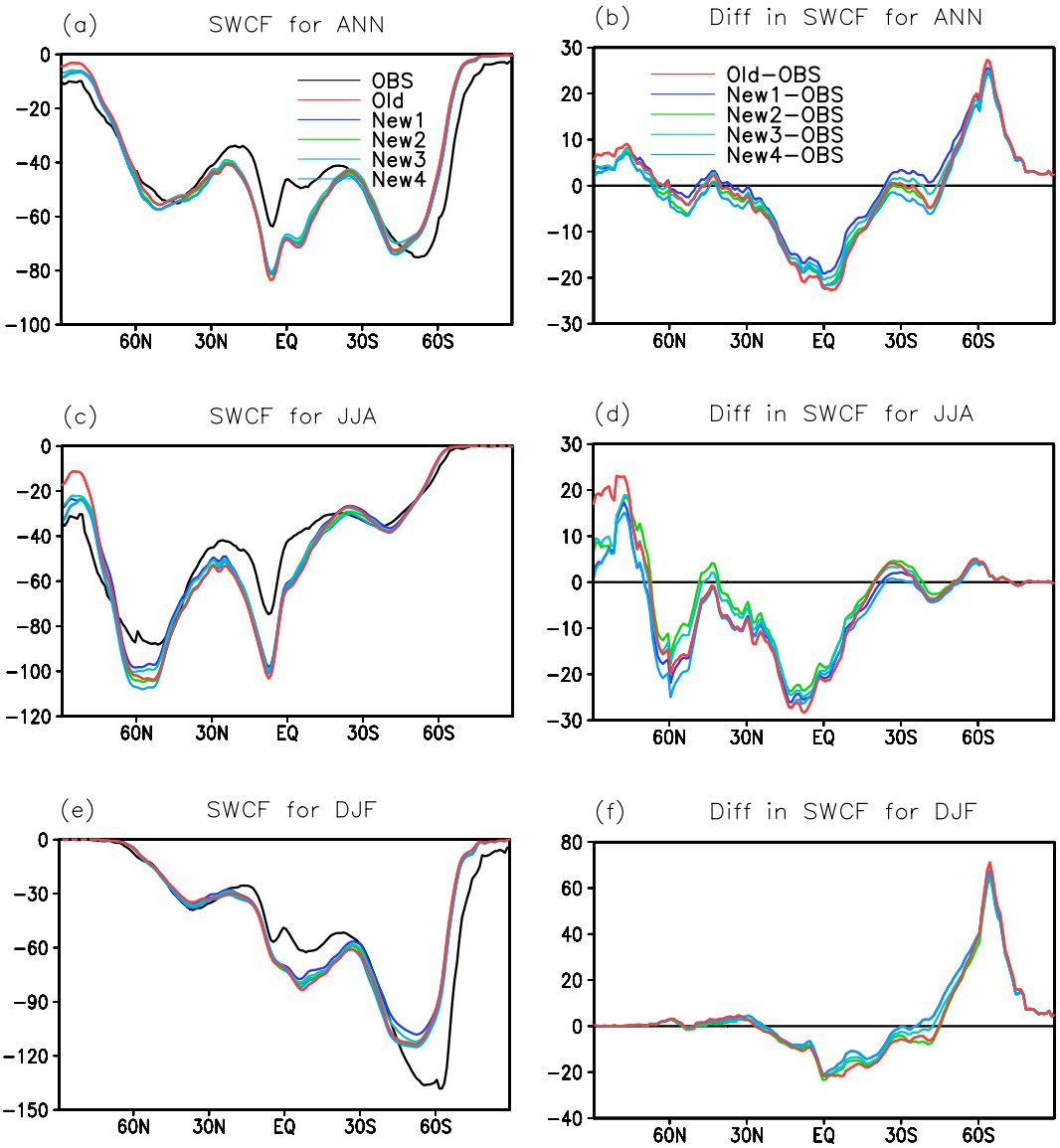

**Figure 2.** Annual, JJA and DJF zonal mean of shortwave cloud radiative forcing (SWCF, W m$^{-2}$) derived from CAM5.1 (a, c and e) and the CERES-EBAF observations (OBS), and their difference between OBS and Old, as well as News (b, d and f).

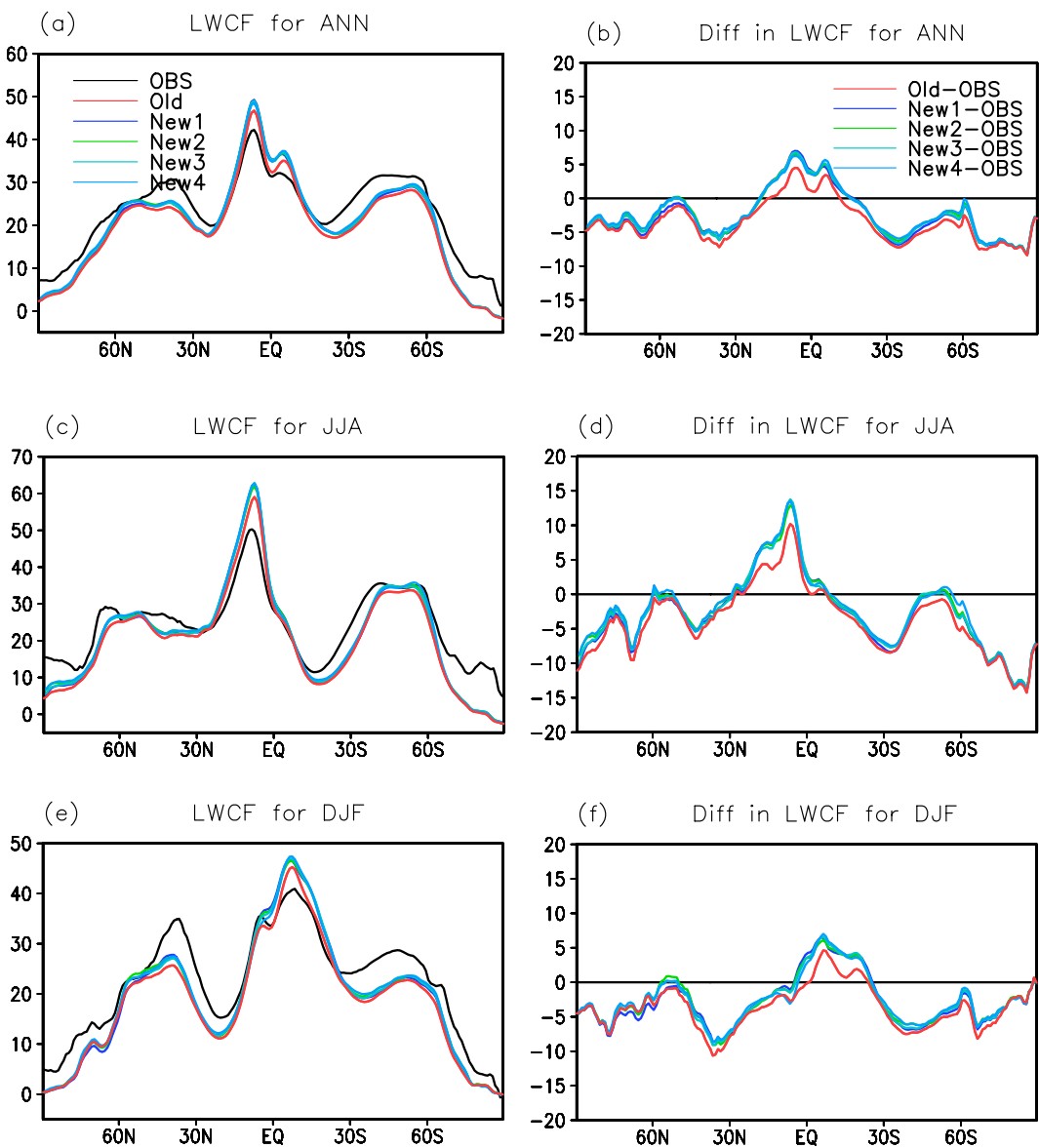

**Figure 3.** Annual, JJA and DJF zonal mean of longwave cloud radiative forcing (LWCF, W m$^{-2}$) derived from CAM5.1 (a, c and e) and the CERES-EBAF observations (OBS), and their difference between OBS and Old, as well as News (b, d and f).

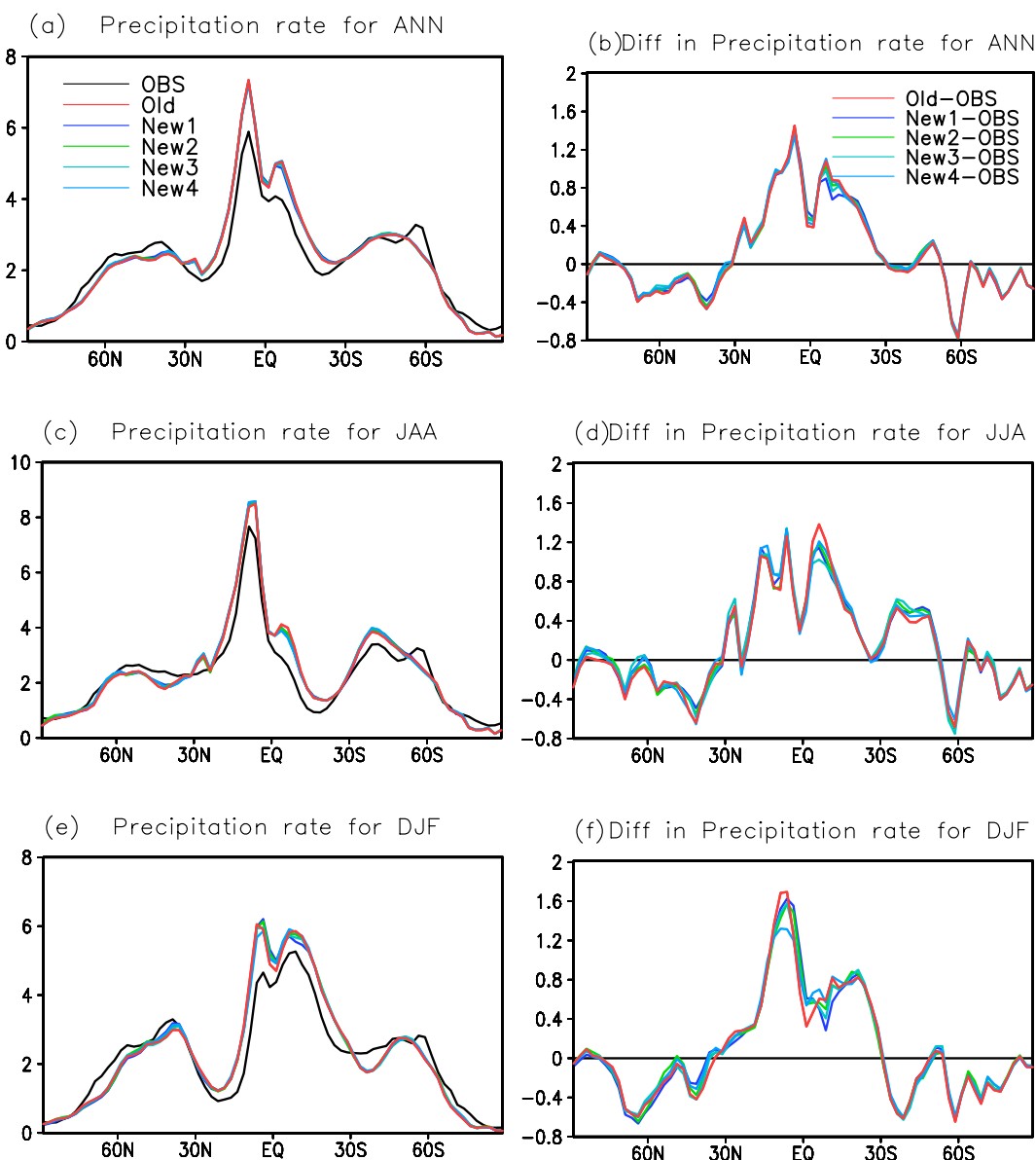

**Figure 4.** Annual, JJA and DJF zonal mean of the total precipitation rate (mm day$^{-1}$) derived from CAM5.1 (a, c and e) and the GPCP observations (OBS), and their corresponding difference between OBS and Old, as well as News (b, d and f).

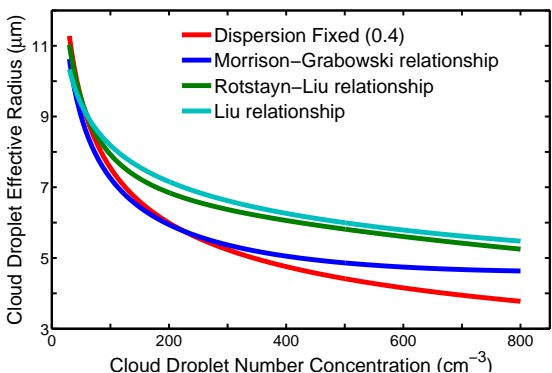

**Figure 5.** Variations in the cloud droplet effective radius as the functions of droplet concentration for relative dispersion fixed at 0.4 (red curve), the Morrison-Grabowski relationship (blue curve), the Rotstayn-Liu relationship (green curve), and the Liu relationship with fixed liquid water content $L_c$ as 0.12 g m$^{-3}$ (cyan curve).

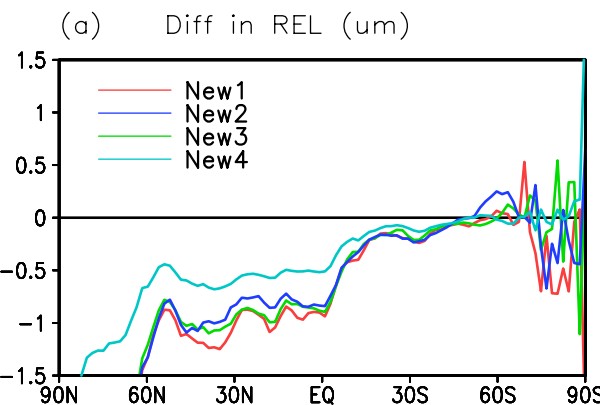

**Figure 6.** Annual zonal mean differences in the cloud top effective radius (REL, $u$m) between PD and PI derived from News.

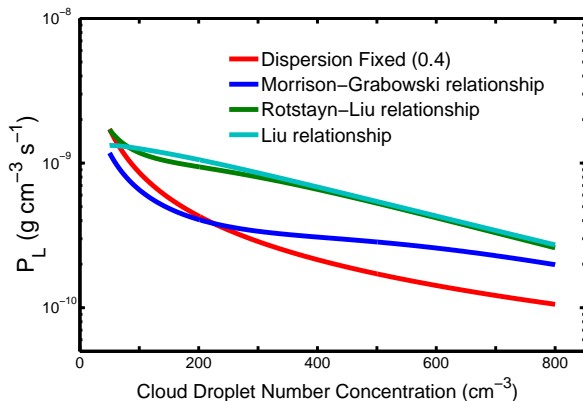

**Figure 7.** Variations in autoconversion rate of the cloud water mass content ($P_L$) as the functions of droplet concentration for Dispersion fixed (0.4), the Morrison-Grabowski relationship, the Rotstayn-Liu relationship, and the Liu relationship ($L_c$ is fixed as 1.4 g m$^{-3}$).

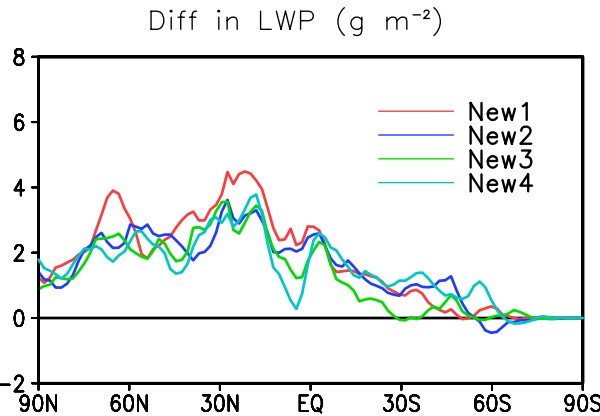

**Figure 8.** Annual, JJA and DJF zonal mean differences in the liquid water path (LWP, g m$^{-2}$) between PD and PI derived from News.

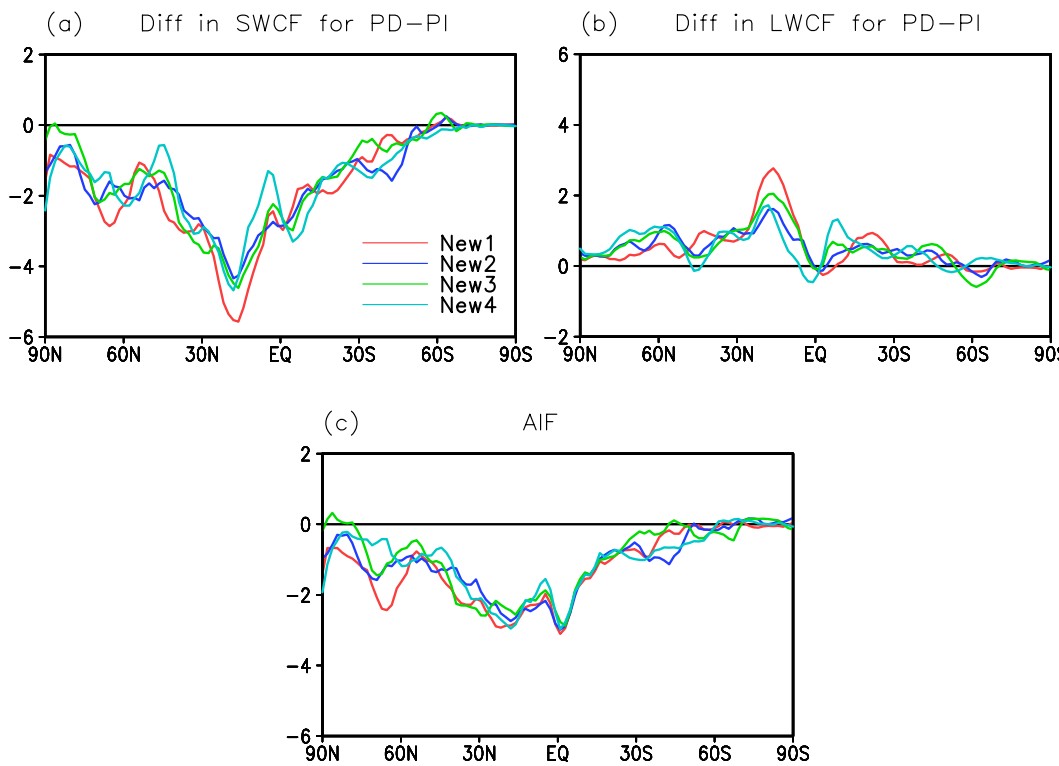

**Figure 9.** Annual, JJA and DJF zonal mean differences in shortwave (SWCF, W m$^{-2}$) and longwave cloud radiative forcing (LWCF, W m$^{-2}$) between PD and PI, as well as aerosol indirect forcing (AIF, W m$^{-2}$) derived from News.