# Peer review of "Sensitivity study of cloud parameterizations with relative dispersion in CAM5.1: impacts on aerosol indirect effects"

_Atmospheric Chemistry and Physics, 2016_

## Referee Comment (RC1) · Anonymous Referee #1 · 7 Feb 2017

General comments: This study examines the behavior of different microphysics schemes used in climate models that take into account the relative dispersion effect in different ways, and explores the sensitivity of the model-simulated cloud and radiation fields to different representations of the dispersion enhancement with increasing aerosols. The results show that the aerosol indirect forcing becomes reduced significantly when incorporating the aerosol-induced increase of the relative dispersion. It is also shown that the reduced magnitude of the indirect forcing depends on choice of the scheme with different sensitivities of the dispersion to droplet number concentration. This is a useful addition to estimates of the aerosol indirect effect, particularly by means of climate modeling. The study is also (at least qualitatively) consistent with a

growing body of knowledge that tends to indicate that the aerosol indirect forcing might be smaller than what has been considered in the past. The important contribution of this study, I think, is a quantitative estimate of how much aerosol indirect forcing can be reduced by the relative dispersion effect. I would recommend the paper be accepted for publication in Atmos. Chem. Phys. after my following concerns are adequately addressed.

Specific points: Page 2, Line 19-20: "$\varepsilon$ is increased by anthropogenic aerosols under similar dynamical conditions in clouds." Why does the relative dispersion increase with increasing cloud droplet number concentration? Please explain the basic mechanisms for it, not just providing a reference to previous studies that showed such tendencies.

Page 5, Line 21-23: "The difference between the simulations with the same ocean surface conditions but aerosol emissions for PD and PI was used to calculate the changes in cloud microphysical properties and cloud radiative forcing induced by anthropogenic aerosols in Section 4." It seems that the aerosol indirect radiative forcing (AIF) thus obtained is the effective radiative forcing that is a "net" radiative forcing remaining after the rapid adjustment occurs, rather than an instantaneous radiative forcing. Is this correct? If so, the authors should clarify that this is the effective radiative forcing, not the instantaneous radiative forcing, because these two are remarkably different in their representations as a climate driver (IPCC AR5, Chapter 7). Even in that case, the reviewer is a bit confused by the author's definition of the indirect radiative forcing (AIF): To the reviewer's understanding, the first indirect effect is categorized into the instantaneous radiative forcing while the second indirect effect is categorized into the effective radiative forcing. The authors, however, tend to define the first and second indirect forcings due to perturbations to Reff and LWP, respectively, in the same configuration of the prescribed SST. Should I interpret the AIF as the total effective radiative forcing due to aerosol-induced perturbation to clouds? I would much appreciate the reviewer to clarify these points.

Page 6, Last paragraph: It is shown that the cloud droplet number concentration is

underestimated while the effective radius agrees with satellites. How should I interpret these apparently inconsistent results? – Does this mean that the cloud water content is also underestimated?

Page 7, Line 8-11: Can these biases in SWCF and LWCF be interpreted in terms of biases in occurrence of different heights of clouds (low, middle and high clouds)? It would be useful to show cloud cover for low, middle and high clouds, as well as the total cloud cover, in Table 2.

Minor points: I found some grammatical errors/typos as follows. Hope this helps the authors improve English.

Page 5, Line 4: "as detailedly described by Neale et al. (2010)" -> "which is documented in Neale et al. (2010)".

Page 5, Line 13: "here" -> "where"

Page 9, Line 13: "PL on Nc" -> "PL with increasing Nc".

Page 9, Line 20: These results can also *be* seen. . .

---

## Referee Comment (RC2) · Anonymous Referee #2 · 10 Feb 2017

The study carried out by Xie et al. implemented a new relative dispersion treatment in the CAM5 cloud parameterization, accounted for its effect of on autoconversion process, and assessed its impact on the climate and aerosol indirect forcing. While this study is suitable for ACP, I have some concerns for the authors to consider when they revise the manuscript.

1. The title: I am not sure if the new relative dispersion treatment constitutes a "New cloud parameterization". I am also not convinced that this study has done enough to be categorized as a "model evaluation" paper as shown in the title since only global means, seasonal means, and zonal means are compared with standardized observational data products. I think this study is a model sensitivity study and the title should

reflect that.

2. The results show that the AIF reduces by only 0.1-0.2W/m2 in CAM5, and this reduction is very small. This is much smaller than the previous study Rotstayn and Liu (2005), which implemented the same relative dispersion representation in the CSIRO Mark3 GCM. It will be interesting to discuss the difference between these two studies.

3. The treatment of dispersion effect on cloud droplet effective radius in the default MG microphysics scheme in CAM5 is based on Morrison and Grabowski (2007) and the new treatment used in this study is based on an earlier study Rotstayn and Liu (2003). I think it might be interesting to discuss why these two formulae are different (e.g., are they based on observations of different cloud regimes?) and provide a justification of your choice of the scheme.

4. Regarding the reference, I think the authors should try to cite other relevant studies on this subject in addition to their own previous studies, especially when the authors use strong wordings such as "it is well established. . .".

---

## Author Comment (AC1) · 28 Mar 2017

Response to Reviewer #1:

General comments:

This study examines the behavior of different microphysics schemes used in climate models that take into account the relative dispersion effect in different ways, and explores the sensitivity of the model-simulated cloud and radiation fields to different representations of the dispersion enhancement with increasing aerosols. The results show that the aerosol indirect forcing becomes reduced significantly when incorporating the aerosol-induced increase of the relative dispersion. It is also shown that the reduced magnitude of the indirect forcing depends on choice of the scheme with different sensitivities of the dispersion to droplet number concentration. This is a useful addition to estimates of the aerosol indirect effect, particularly by means of climate modeling. The study is also (at least qualitatively) consistent with a growing body of knowledge that tends to indicate that the aerosol indirect forcing might be smaller than what has been considered in the past. The important contribution of this study, I think, is a quantitative estimate of how much aerosol indirect forcing can be reduced by the relative dispersion effect. I would recommend the paper be accepted for publication in Atmos. Chem. Phys. after my following concerns are adequately addressed.

Response: Thanks for the positive comments.

Specific points:

Page 2, Line 19-20: "$\varepsilon$ is increased by anthropogenic aerosols under similar dynamical conditions in clouds." Why does the relative dispersion increase with increasing cloud droplet number concentration? Please explain the basic mechanisms for it, not just providing a reference to previous studies that showed such tendencies.

Yes, we have added the explanations in the revised manuscript: *"Liu and Daum (2002) suggested that $\varepsilon$ is increased by anthropogenic aerosols under similar dynamical conditions in clouds, because more numerous small droplets formed in polluted clouds compete for water vapor and broaden the droplet size distribution compared with clean clouds having fewer droplets and less competition. Further theoretical*

*study (Liu et al., 2006) revealed that the increased ε is primarily due to slowdown of condensational narrowing associated with decreased supersaturation."*

Page 5, Line 21-23: "The difference between the simulations with the same ocean surface conditions but aerosol emissions for PD and PI was used to calculate the changes in cloud microphysical properties and cloud radiative forcing induced by anthropogenic aerosols in Section 4." It seems that the aerosol indirect radiative forcing (AIF) thus obtained is the effective radiative forcing that is a "net" radiative forcing remaining after the rapid adjustment occurs, rather than an instantaneous radiative forcing. Is this correct? If so, the authors should clarify that this is the effective radiative forcing, not the instantaneous radiative forcing, because these two are remarkably different in their representations as a climate driver (IPCC AR5, Chapter 7). Even in that case, the reviewer is a bit confused by the author's definition of the indirect radiative forcing (AIF): To the reviewer's understanding, the first indirect effect is categorized into the instantaneous radiative forcing while the second indirect effect is categorized into the effective radiative forcing. The authors, however, tend to define the first and second indirect forcings due to perturbations to Reff and LWP, respectively, in the same configuration of the prescribed SST. Should I interpret the AIF as the total effective radiative forcing due to aerosol-induced perturbation to clouds? I would much appreciate the reviewer to clarify these points.

Thanks for this great point. In our manuscript, the combined first and second indirect forcing is the effective radiative forcing, not instantaneous radiative forcing. The AIF is the combined first and second indirect forcing, which is the total effective radiative forcing due to aerosol-induced perturbation to clouds including the first and second indirect effects. Hence, we have clarified in our manuscript *"Note that the AIF is the combined first and second indirect forcing, which is the effective radiative forcing (net TOA radiative fluxes to perturbations with rapid adjustments), not instantaneous radiative forcing, following IPCC (2013)."*

Page 6, Last paragraph: It is shown that the cloud droplet number concentration is underestimated while the effective radius agrees with satellites. How should I interpret these apparently inconsistent results? – Does this mean that the cloud water content is also underestimated?

Yes, "the simulated cloud droplet number concentration is underestimated in CAM5.1 model while the effective radius agrees with satellites" is right. This apparent insistency could arise from underestimated cloud water content. Unfortunately, we do not have observed cloud water content to verify this point (Gettelman et al., 2015). To accommodate this point, we add in revision *"It is noted that the simulated cloud droplet number concentration is underestimated in CAM5.1 model while the effective radius agrees with satellites. This apparent inconsistency suggests that the simulated liquid water content may be somehow underestimated. Unfortunately, we do not have observed cloud water content to verify this (Gettelman et al., 2015)."*

Page 7, Line 8-11: Can these biases in SWCF and LWCF be interpreted in terms of biases in occurrence of different heights of clouds (low, middle and high clouds)? It would be useful to show cloud cover for low, middle and high clouds, as well as the total cloud cover, in Table 2.

Thanks for the suggestions. We have added the statistical properties for low, middle and high clouds at the global scale in Table 2, and we also have added some interpretations about SWCF and LWCF in the terms of low, middle and high clouds in the revised manuscript.

**Table 2.** Annual global mean aerosols, cloud properties, and surface precipitation, as well as TOA energy budget with year 2000 aerosol emissions including aerosol optical depth at wavelength 550 nm (AOD), liquid water path (LWP), Ice water path (IWP), the vertical integrated cloud droplet number concentration ($N_d$), cloud top effective radius (REL), total cloud fraction (CLDTOT), low cloud fraction (CLDLOW), middle cloud fraction (CLDMED), high cloud fraction (CLDHGH), total precipitation rate ($P_{tot}$), shortwave cloud radiative forcing (SWCF), and longwave cloud radiative forcing (LWCF).

| Simulation | Old | New1 | New2 | New3 | New4 | OBS |
|---|---|---|---|---|---|---|
| AOD | 0.121 | 0.122 | 0.122 | 0.124 | 0.125 | 0.15[a] |
| LWP, g m$^{-2}$ | 44.74 | 36.76 | 40.33 | 37.62 | 43.48 | – |
| IWP, g m$^{-2}$ | 17.78 | 18.70 | 18.88 | 18.84 | 18.96 | – |
| $N_d$, $10^{10}$ m$^{-2}$ | 1.38 | 1.33 | 1.40 | 1.35 | 1.47 | 4.01[b] |
| REL, $\mu$m | 9.21 | 11.48 | 10.87 | 11.32 | 10.08 | 10.5[b] |
| CLDTOT, % | 64.02 | 65.50 | 65.63 | 65.74 | 65.82 | 65–75[c] |
| CLDLOW, % | 43.61 | 44.88 | 45.25 | 45.31 | 45.47 | – |
| CLDMID, % | 27.27 | 27.58 | 27.67 | 27.65 | 27.72 | – |
| CLDHGH, % | 38.09 | 39.24 | 39.09 | 39.22 | 39.16 | 21–33[d] |
| $P_{tot}$, mm day$^{-1}$ | 2.96 | 2.97 | 2.97 | 2.97 | 2.97 | 2.67[e] |
| SWCF, W m$^{-2}$ | −52.08 | −49.82 | −52.40 | −51.01 | −53.03 | −47.07[f] |
| LWCF, W m$^{-2}$ | 24.06 | 25.23 | 25.40 | 25.37 | 25.51 | 26.48[f] |

[a] AOD is from a satellite retrieval composite (Kinne et al., 2006), [b] $N_d$ and REL are from the AVHRR data (Han et al., 1998). [c] CLDTOT is obtained from ISCCP (Rossow and Schiffer, 1999), MODIS data (Platnick et al., 2003), and HIRS data (Wylie et al., 2005). [d] CLDHGH is obtained from ISCCP data (Rossow and Schiffer, 1999) and HIRS data (Wylie et al., 2005). [e] $P_{tot}$ is taken from the Global Precipitation Climatology Project (GPCP) for the years 1979−2009 (Adler et al., 2003). [f] Radiative fluxes from the CERES-EBAF are for the years 2000−2010 from Loeb et al. (2009).

Minor points: I found some grammatical errors/typos as follows. Hope this helps the authors improve English.

Thank you very much for your kindness. We have checked it and tried our best to correct all the errors/typos.

Page 5, Line 4: "as detailedly described by Neale et al. (2010)" -> "which is documented in Neale et al. (2010)".

Taken.

Page 5, Line 13: "here" -> "where"

Taken.

Page 9, Line 13: "PL on Nc" -> "PL with increasing Nc".

Taken.

Page 9, Line 20: These results can also *be* seen: : :

Taken.

References

Gettelman, A., Morrison, H., Santos, S., Bogenschutz, P., and Caldwell, P. M.: Advanced two-moment bulk microphysics for global models. Part II: global model solutions and aerosol-cloud interactions, J. Climate, 28, 1288–1307, doi:10.1175/JCLI-D-14-00103.1, 2015.

IPCC: Climate Change 2013: The Physical Science Basis. Contribution of Working Group I to the Fifth Assessment Report of the Intergovernmental Panel on Climate Change, edited by: Stocker, T. F., Qin, D., Plattner, G.-K., Tignor, M., Allen, S. K., Boschung, J., Nauels, A., Xia, Y., Bex, V., and Midgley, P. M., Cambridge University Press, Cambridge, United Kingdom and New York, NY, USA, 1535 pp., 2013

Liu, Y. and Daum, P. H.: Indirect warming effect from dispersion forcing, Nature, 419, 580–581, 2002.

Liu, Y., Daum, P. H., and Yum, S. S.: Analytical expression for the relative dispersion of the cloud droplet size distribution, Geophys. Res. Lett., 33, L02810, doi:10.1029/2005GL024052, 2006.

---

## Author Comment (AC2) · 28 Mar 2017

Response to Reviewer #2:

General comments:

The study carried out by Xie et al. implemented a new relative dispersion treatment in the CAM5 cloud parameterization, accounted for its effect of on autoconversion process, and assessed its impact on the climate and aerosol indirect forcing. While this study is suitable for ACP, I have some concerns for the authors to consider when they revise the manuscript.

Response: Thank the Reviewer very much for the comments.

Specific comments:

1. The title: I am not sure if the new relative dispersion treatment constitutes a "New cloud parameterization". I am also not convinced that this study has done enough to be categorized as a "model evaluation" paper as shown in the title since only global means, seasonal means, and zonal means are compared with standardized observational data products. I think this study is a model sensitivity study and the title should reflect that.

Thanks for the suggestion. We have changed the title and the new title is "*Sensitivity study of cloud parameterizations with relative dispersion in CAM5.1: model evaluation and impacts on aerosol indirect effects.*" Furthermore, we compared key statistical measures based on global spatial distribution including spatial pattern correlation and the root mean squared error with observed data products in Table 2 (SWCF), Table 3 (LWCF), and Table 4 (precipitation rate), in addition to comparing the global means, seasonal means, and zonal means.

2. The results show that the AIF reduces by only 0.1-0.2W/m2 in CAM5, and this reduction is very small. This is much smaller than the previous study Rotstayn and Liu (2005), which implemented the same relative dispersion representation in the CSIRO Mark3 GCM. It will be interesting to discuss the difference between these two studies.

Thanks for pointing this out. The reduction of AIF in our model is much smaller than

that from Rotstayn and Liu (2005). We think that a main reason is that the reference cases are different. In Rotstayn and Liu (2005), the reference case is performed with the autoconversion parameterization (Baker, 1993; Boucher et al., 1995) given below,

$$P = E_c \pi \kappa_1 \left( \frac{3}{4\pi\rho_l} \right)^{4/3} N^{-1/3} L^{7/3} H(R_3 - R_{3c}),$$

In our case, the reference autoconversion parameterization is

$$P_N = 1.1 \times 10^{10} \frac{\Gamma(\varepsilon^{-2}, x_{cq})\Gamma(\varepsilon^{-2}+6, x_{cq})}{\Gamma^2(\varepsilon^{-2}+3)} L_c{}^2,$$

$$P_L = 1.1 \times 10^{10} \frac{\Gamma(\varepsilon^{-2})\Gamma(\varepsilon^{-2}+3, x_{cq})\Gamma(\varepsilon^{-2}+6, x_{cq})}{\Gamma^3(\varepsilon^{-2}+3)} N_c{}^{-1} L_c{}^3,$$

with fixed dispersion of 0.4. According to the Reviewer's suggestions, we have added some discussions in revision: *"It is worth noting that the reduction of AIF induced by dispersion effect in this study is much smaller than that (approximately −0.5 W m$^{-2}$ for global means) reported by Rotstayn and Liu (2005). This difference lies likely in the reference autocnversion parameterations. In this study, Eq. (3) with fixed dispersion of 0.4 is used whereas Rotstayn and Liu (2005) used a different one give in*

$$P_L = E_c \pi \kappa_1 (\tfrac{3}{4\pi\rho_l}) N^{-1/3} L^{7/3} H(R_3 - R_{3c}). \text{ ,,}$$

3. The treatment of dispersion effect on cloud droplet effective radius in the default MG microphysics scheme in CAM5 is based on Morrison and Grabowski (2007) and the new treatment used in this study is based on an earlier study Rotstayn and Liu (2003). I think it might be interesting to discuss why these two formulae are different (e.g., are they based on observations of different cloud regimes?) and provide a justification of your choice of the scheme.

Thanks for pointing this out. We have added some discussions in revision: *"The Morrison-Grabowski relationship is based on small number of measurements (ε=0.33 for maritime air masses and ε=0.43 for continental air masses) reported in Martin et al., 1994, while the Rotstayn and Liu relationship is derived from more measurements described by Liu and Daum (2002). Also, the Rotstayn-Liu relationship assumes the dispersion levels off at approximately 800 cm$^{-3}$ while the linear Morrison-Grabowski*

*relationship has no such limit."*

4. Regarding the reference, I think the authors should try to cite other relevant studies on this subject in addition to their own previous studies, especially when the authors use strong wordings such as "it is well established: : :".

Thank you for your good suggestions about adding other relevant studies. In the paragraph, we have added some important references. Hence, the sentence has been modified as *"It is well established that effective radius (Martin et al; 1994; Liu and Daum, 2002) and autoconversion rate (Liu and Daum, 2004; Liu et al., 2007; Xie and Liu, 2009; Li et al., 2008; Chuang et al. 2012; Wang et al., 2013; Michibata and Takemura, 2015) are both related to the relative dispersion of cloud droplet size distribution ε (which is defined as the ratio of the standard deviation to the mean value of droplet size distribution) in addition to droplet number concentration and cloud liquid water content."*

References

Baker, M. B.: Variability in concentrations of cloud condensation nuclei in the marine cloud-topped boundary layer, Tellus, Ser. B, 45, 458–472, 1993.

Boucher, O., Le Treut, H., and Baker, M. B.: Precipitation and radiation modelling in a GCM: Introduction of cloud microphysical processes, J. Geophys. Res., 100, 16,395–16,414, 1995.

Liu, Y. and Daum, P. H.: Indirect warming effect from dispersion forcing, Nature, 419, 580–581, 2002.

Martin, G. M., Johnson, D. W., and Spice, A.: The measurement and parameterization of effective radius of droplets in warm stratocumulus clouds, J. Atmos. Sci., 51, 1823–1842, 1994.

Rotstayn, L. D. and Liu, Y.: A smaller global estimate of the second indirect aerosol effect, Geophys. Res. Lett., 32, L05708, doi:10.1029/2004GL021922, 2005.

---

## Author Response (AR2)

Manuscript No: **acp-2016-1172**

Journal: **ACP**

The revised manuscript entitled "**Sensitivity study of cloud parameterizations with relative dispersion in CAM5.1: impacts on aerosol indirect effects**" by Xiaoning Xie, He Zhang, Xiaodong Liu, Yiran Peng, and Yangang Liu.

We thank the editor and the second Reviewer for their two suggestions to further improve our manuscript. We have addressed the two concerns, with point-by-point responses detailed below (reviewers comments in red color and our responses in blue color). Based on the second Reviewer's suggestion, we have changed the title as "Sensitivity study of cloud parameterizations with relative dispersion in CAM5.1: impacts on aerosol indirect effects".

Response to the editor and the second Reviewer:

The paper is now almost acceptable, but please do follow the remaining two suggestions by the reviewer and refine the title, and add some more discussion about the difference to the Rotstayn and Liu (2005) results.

Best regards,

Johannes

Thank the editor for his positive comments. In the following section, we have modified our manuscript according to these two valuable suggestions.

The authors have addressed most of my comments, but I have a couple of minor comments on this manuscript.

1. Regarding the title and the scope of this study: I still do not think this is a "model evaluation" paper since it only provides some very basic comparison between model simulations and observations. I think the scope of this paper is to test the sensitivity of the model to relative dispersion representations, and the title in the revised version is better than the previous one but still needs to be further refined.

Yes, we have deleted the "model evaluation" in our title according to the second Reviewer's suggestion. Hence, we have changed the title as "Sensitivity study of cloud parameterizations with relative dispersion in CAM5.1: impacts on aerosol indirect effects".

2. Regarding my previous comments on reconciling the differences between this study and Rotstayn and Liu (2005), I would encourage the authors to elaborate more and explain the physical processes that drive the differences.

[revised manuscript text omitted]